# TRANSFER VALUE OR POLICY?
# A VALUE-CENTRIC FRAMEWORK TOWARDS TRANSFERRABLE CONTINUOUS REINFORCEMENT LEARNING

## ABSTRACT

Transferring learned knowledge from one environment to another is an important step towards practical reinforcement learning (RL). In this paper, we investigate the problem of transfer learning across environments with different dynamics while accomplishing the same task in the continuous control domain. We start by illustrating the limitations of policy-centric methods (policy gradient, actor-critic, etc.) when transferring knowledge across environments. We then propose a general model-based value-centric (MVC) framework for continuous RL. MVC learns a dynamics approximator and a value approximator simultaneously in the source domain, and makes decision based on both of them. We evaluate MVC against popular baselines on 5 benchmark control tasks in a training from scratch setting and a transfer learning setting. Our experiments demonstrate MVC achieves comparable performance with the baselines when it is trained from scratch, while it significantly surpasses them when it is used in the transfer setting.

## 1 INTRODUCTION

While the achievements of deep reinforcement learning (DRL) are exciting in having conquered many computer games (Go (Silver et al., 2016; 2017), Atari games (Mnih et al., 2015)), in practice, it is still hard for these algorithms to find applications in real-world environments. Among the impediments, a primary obstacle is that they could fail even if the environment to deploy the agent is slightly different from where they were trained (Tan et al., 2018; Bousmalis et al., 2018; Tobin et al., 2017; Tamar et al., 2016). In other words, they lack the desired ability to *transfer* experiences learned from one environment to another. These observations motivate us to ask, *(1) why are current RL algorithms so inefficient in transfer learning and (2) what kind of RL algorithms could be friendly to transfer learning by nature?*

In this work, we explore the two questions and present a partial answer based on analysis and experiments. Our exploration concentrates on control tasks due to its broad impact; in particular, we further assume that across environments only their dynamics are not the same. Possible sources of such dynamics discrepancy could be variation of physical properties such as object mass, gravity, and surface friction. It is worth noting that our framework is general and we do not assume any specific perturbation source or type.

Our investigation starts with understanding the limitation of transferring the policy function (which maps a state to a distribution of actions) across environments. We analyze this transfer strategy because the de facto DRL framework in the control domain (DDPG (Lillicrap et al., 2015), TRPO (Schulman et al., 2015), A3C (Mnih et al., 2016), etc.) are policy-centric methods, which directly optimize the policy function. As a result, these methods learn a *precise* policy function, and sometimes also produce an *imprecise* value/Q-function as a side product. However, even if a perfect policy function has been learned from the source environment, this policy could behave quite poorly, or even fail, in the new environment, especially when the action space is hard-constrained (e.g., force or torque usually has a maximal value). We illustrate this by a simple example: Imagine a child shooting three-pointers when playing basketball. With a 600g ball, she can make the three-pointer. However, she may hardly make the three with a 800g ball because it is too heavy. What will she do? Most likely she will step forward, approach the basket, and make a closer jump-shot. We see that *marginal*

*dynamics variation can lead to drastic policy change*, and direct policy optimization initialized from the old policy would not be efficient. We will analyze this issue more systematically by theoretical and experimental approaches throughout the paper.

The investigation implies that, instead of directly transferring policies, the swift transfer should be grounded in richer and more structured knowledge of the task, so as to facilitate the judgment of whether the agent is approaching the goal, which is critical for making the right decision. Enlightened by the above intuition, we propose a simple model-based and value-centric framework for continuous reinforcement learning.

Our method contains two disentangled components: a dynamics approximator (model-based) and a state value function approximator (value-centric). The agent plans its action by solving an optimization problem using both approximators. As side products, this design learns a precise transition function, a precise reward function, and a precise value function on a subset of states. In particular, knowledge from historical explorations have been stored in the value function. In comparison, previous policy-centric methods can only produce a precise policy function, thus our framework allows to transfer much more information. By fine-tuning the whole framework in a new environment, our agent can adapt quickly with much lower sample complexity than state-of-the-art.

We call our method value-centric because it strives to learn a *precise* value function. The general framework is inspired from the Value Iteration (VI) method, which is a classical approach for discrete decision making. However, since control problems have a *continuous action space*, we cannot directly enumerate over the action space as in the discrete setting but have to address the highly non-convex optimization problem. To make it tractable, we leverage differentiable function approximators like neural networks to learn the dynamics and the value function. By such an approximation, it is possible to solve the optimization problem with state-of-the-art optimizers effectively.

We also theoretically analyze our value-centric framework and classical policy gradient algorithms from an optimization perspective. To build an intuitive understanding, we create a simple and illustrative example that clearly shows a local optimum in the policy space can prevent policy gradient methods from transferring successfully, but will not affect our value-centric framework.

We summarize our contributions as below:

- We provide a theoretical justification to show the advantage of value-centric methods from an optimization perspective.

- We propose a novel value-centric framework for continuous reinforcement learning of comparable sample efficiency with popular deep RL methods in the training from scratch setting.

- Extensive experiments show the superiority of our method in a transfer learning setting

## 2 BACKGROUND

### 2.1 MDPS AND NOTATION

First, we introduce the standard reinforcement learning setting for continuous control. We assume the underlying control problem is a *Markov Decision Process* (MDP), which is defined by a tuple $\langle \mathcal{S}, \mathcal{A}, \mathcal{T}, \mathcal{R}, \rho_0, \gamma \rangle$. Here, $\mathcal{S}$ and $\mathcal{A}$ are the continuous state space and action space, respectively, $\mathcal{T}$ is the transition dynamics, $\mathcal{R}$ is the reward function, $\rho_0$ is the distribution of the initial state, and $\gamma \in (0, 1]$ is the discounting factor for future rewards. For robot control tasks, it is reasonable to assume a deterministic transition dynamics $\mathcal{T} : \mathcal{S} \times \mathcal{A} \rightarrow \mathcal{S}$, and a deterministic reward function $\mathcal{R} : \mathcal{S} \times \mathcal{A} \rightarrow \mathbb{R}$.

The policy $\pi$ of an agent maps the state space to a probability distribution over the action space $\pi : \mathcal{S} \rightarrow \mathcal{P}(\mathcal{A})$. The goal of reinforcement learning is to find an optimal policy $\pi^*$ that maximizes the expectation of the accumulated future rewards according to the initial state distribution $J(\pi) = E_{\rho_0, \pi}[\sum_{t=0}^{\infty} \gamma^t \mathcal{R}(s_t, a_t)]$. We also define the value function $V$ of a state $s$ as $V(s) = E_\pi[\sum_{t=0}^{\infty} \gamma^t \mathcal{R}(s_t, a_t)|s_0 = s]$. Finally, the optimal value function $V^*$ satisfies $V^*(s) = \max_\pi E_\pi[\sum_{t=0}^{\infty} \gamma^t \mathcal{R}(s_t, a_t)|s_0 = s]$ for any $s \in \mathcal{S}$.

**Low-dimensional Assumption** We focus on control problems which usually have well-engineered and low-dimensional state/action representations. Not rigorously, the assumption has two implications:

- **Property 1:** For a smooth function $f(a_t)$ over $\mathcal{A}$, we can find its approximate solution by sampling over the domain and optimizing locally;
- **Property 2:** We can learn a network to approximate the transition and reward functions.

Empirically, we find evidence of both properties as in our experiment section (Sec 6).

## 2.2 TRANSFER LEARNING

Many differently posed transfer learning problems have been discussed in the reinforcement learning literature (Taylor & Stone, 2009). In this work, we study the problem of the environment slightly changing while the task remains the same. For example, in the pendulum swing-up problem, once the agent learns how to swing up a 1kg pendulum, we expect that it could quickly adapt itself to swing up a 2kg pendulum leveraging the learned knowledge. We formulate our setting by modifying the aforementioned standard RL setting. We consider a pair of MDPs sharing state and action spaces. Their transition dynamics $\mathcal{T}$ and reward functions $\mathcal{R}$ are parameterized by a vector $\xi$: $s_{t+1} = \mathcal{T}(s_t, a_t; \xi), r_t = \mathcal{R}(s_t, a_t; \xi)$. Each $\xi$ defines a unique MDP $\mathcal{M}_\xi$. The change of $\mathcal{R}$ is only caused by the change of $\mathcal{T}$ (instead of the goal of the task), so the change of reward function is limited, though we parameterize it by $\xi$ for rigor. After the agent learns how to perform well on a source MDP, we expect it to solve the target MDP using few interactions with the new environment.

## 3 WHY TRANFERRING POLICY CAN BE DIFFICULT? A SIMPLE ILLUSTRATION

In the RL community, most control tasks are solved by policy gradient-based methods. In this section, we illustrate the limitation of transferring policy by a simple example and compare it with a value-centric method.

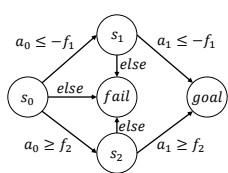

Figure 1: an MDP

We design an MDP with 5 states as in Fig. 1: an initial state ($s_0$), two intermediate states ($s_1, s_2$), a goal state ($s_g$), and a failure state ($s_f$). $s_g$ and $s_f$ are terminal states, so their state values are both 0. The task for the agent is to reach the goal state through one of the two paths. We denote the path passing $s_1$ as $path_1$, and the path passing $s_2$ as $path_2$. Let $f_i > 0$ be environment parameters. At some state $s$, the agent takes an action $a \in [-1, 1]$. The reward is designed in a goal-based style with penalties to large $|a|$. If the agent transits from the initial state to an intermediate state, it receives a reward $r = 1 - |a|$. Every time the agent visits the failure state, it will be punished by a negative reward $r = -5$. The optimal policy for this task is $a_0^* = -f_1$ if $f_1 < f_2; a_0^* = f_2$ if $f_1 \geq f_2; a_1^*(s_1) = -f_1; a_1^*(s_2) = f_2$.

Now we show the different behaviors of the value- and policy-centric strategies in a transfer learning setting. For the source environment, $f_1 = 0.7$ and $f_2 = 0.8$. It is not hard to find that the optimal path is $path_1$. Then we modify the transition rule of the environment by slightly varying $f_1$ and $f_2$. We set $f_1 = 0.8$ and $f_2 = 0.7$. While the variation of the environment is relatively small, the optimal policy for $s_0$ is completely different and has changed to $path_2$. The optimal state value function and policy are shown in Table 1 in the Appendix B.

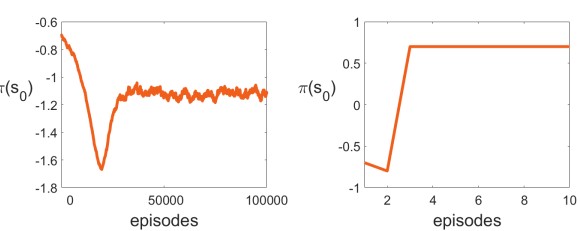

Figure 2: The training curve of $\pi(s_0)$ during fine-tuning. $\pi(s_0)$ represents its mean $\mu(s_0)$ for clarity.

To compare policy- and value-centric strategies, we run two representative algorithms on this game – the Policy Gradient (PG) algorithm (Williams, 1992) (policy-centric) and the Value Iteration (VI) algorithm (Sutton et al., 1998) (value-centric). We assume the value-centric methods have access

to the oracle transition rules and the reward function, and we parameterize the policy $\pi(s_i)$ with Gaussian distribution. Details can be found in Appendix B. We compare the behaviors of PG and VI to intuitively demonstrate their characteristics. PG gets stuck at the old optimum (Fig. 2 left) since the distance between the old optimum and the new optimum is so large. To see why, we first point out that PG optimizes in the policy space, restricting that the policy shifting from the old optimum to the new optimum must be quite "continuous". However, to reach the new optimum, $\pi(s_0)$ must flip the sign of its mean $\mu(s_0)$. Unfortunately, when $\mu(s_0)$ approaches zero from the negative side (from $-0.7$), its transition possibility to the failure state will increase, thus $\mu(s_0)$ will bounce away from $0-$ to avoid punishment. Consequently, it is very unlikely to successfully reach the new optimum, a positive value.

In contrast, the value function in VI algorithm continuously shifts and converges to the new optimum (See Appendix B) and the policy deduced from the value function converges to the optimal policy in 10 episodes. Value function reuses the learned information more efficiently. It also allows sudden change of the policy (Fig. 2 right), which could be beneficial for transferring.

## 4 MODEL-BASED AND VALUE-CENTRIC (MVC) REINFORCEMENT LEARNING

### 4.1 VALUE-CENTRIC METHOD IN TRANSFER LEARNING

Our objective is to make the agent learn faster in a new environment after trained in a similar environment. From the above discussion, we know that instead of directly transferring policies, the swift transfer should be grounded in richer and more structured knowledge of the task. It is worth exploring transfer by value function since the value function contains more information than the policy alone. A straight-forward idea is to utilize the value/Q function from actor-critic algorithms such as DDPG to facilitate transfer learning. However, the value/Q function learned from the actor-critic algorithm is usually imprecise, and it does not capture so much knowledge in the original environment. To address this issue, we propose an algorithm to directly train a precise value function in the original environment. We call it a value-centric method. Then, we just need to fine-tune this value function to help the agent adapt to the new environment. In the following subsections, we explain how to train this precise value function from scratch.

### 4.2 CONTINUOUS VALUE ITERATION

To better explain our main algorithm, we first propose a Continuous Value Iteration algorithm, which is possible to make use of Property 1 in Sec 2.1 to build a value-centric method. Theorem 2 in the Appendix suggests it always converges.

**Definition 1.** *At the $i$-th iteration, we update the value function by the following rule:*

$$V^{i+1}(s) = (1-\alpha)V^i(s) + \alpha(\max_a[\mathcal{R}(s,a,\mathcal{T}(s,a)) + \gamma V^i(\mathcal{T}(s,a))]), \forall s \in \mathcal{S} \quad (1)$$

*where $\alpha \in (0,1]$ is the step size, and $V^0(s)$ is some initialization value. Note that by Property 1 in Sec 2.1, the maximization algorithm over $a$ is accessible. We call this algorithm Continuous Value Iteration.*

### 4.3 ALGORITHM

The CVI algorithm mentioned in Sec 4.2 assumes the oracle of dynamics, rewards, and an algorithm to solve the optimization problem w.r.t. $a$. In addition, it assumes the ability to sweep over the state space. To make it practical, we propose a model-based and value-centric(MVC) framework. We approximate the value function $V$, transition dynamics $\mathcal{T}$, and the reward function $\mathcal{R}$ by function approximators like neural networks. Any differentiable function approximators not restricted to neural networks may be applicable. Let us denote the three parameterized function approximators we use as $f_V(s; \theta), f_\mathcal{T}(s, a; \phi), f_\mathcal{R}(s, a; \psi)$, where $\theta$, $\phi$, and $\psi$ are parameters. To efficiently use the sampled data, we store the agent's experience $(s_t, a_t, r_t, s_{t+1})$ in a replay buffer $\mathcal{B}$. Then $f_\mathcal{T}$ is trained to minimize the following $L_2$ loss function:

$$\mathcal{L}_\mathcal{T} = \frac{1}{|\mathcal{D}|} \sum_{(s_t, a_t, s_{t+1}) \in \mathcal{D}} \|f_\mathcal{T}(s_t, a_t; \phi) - (s_{t+1} - s_t)\|^2 \quad (2)$$

---

**Algorithm 1** Model-based Value-Centric (MVC) Reinforcement Learning

---

**Require:** An MDP $\mathcal{M}$, a maxmimum timestep number $N$
 1: For train from scratch: initialize $f_V(s;\theta), f_{\mathcal{T}}(s,a;\phi), f_{\mathcal{R}}(s,a;\psi)$ randomly.
    For transfer learning: initialize $f_V(s;\theta), f_{\mathcal{T}}(s,a;\phi), f_{\mathcal{R}}(s,a;\psi)$ with the learned parameters.
 2: Initialize an empty replay buffer $\mathcal{B}$.
 3: **for** $i = 1 : N$ **do**
 4:    At state $s_t$, compute action $a_t$ with Eq. 4. Add exploration noise to $a_t$.
 5:    Execute $a_t$ in the environment, receive $r_t$ and $s_{t+1}$. Store $(s_t, a_t, r_t, s_{t+1})$ in $\mathcal{B}$
 6:    Update $f_V(s;\theta)$ with Eq. 5.
 7:    Update $f_{\mathcal{T}}(s,a;\phi), f_{\mathcal{R}}(s,a;\psi)$ with Eq. 2 and Eq. 3
 8:    Soft update target network $f_V(s;\theta^-)$: $\theta^- = \alpha\theta + (1-\alpha)\theta^-$
 9: **end for**

---

Here, $\mathcal{D}$ is the training dataset extracted from $\mathcal{B}$ and $|\cdot|$ refers to the cardinality of a set. Like previous works (Kurutach et al., 2018; Nagabandi et al., 2017), the supervision signal for $f_{\mathcal{T}}$ is the increment of the state to release its burden of memorizing the last state. We also train the reward approximator in a similar supervised learning way:

$$\mathcal{L}_{\mathcal{R}} = \frac{1}{|\mathcal{D}|} \sum_{(s_t,a_t,r_t)\in\mathcal{D}} \|f_{\mathcal{R}}(s_t,a_t;\psi) - r_t\|^2 \tag{3}$$

At time step $t$, the agent samples data under a deterministic policy:

$$\pi(s_t) = \arg\max_a (f_{\mathcal{R}}(s_t,a) + \gamma f_V(f_{\mathcal{T}}(s_t,a) + s_t)) \tag{4}$$

Unlike the tabular case, the right-hand side of Eq. 4 is a complicated non-convex optimization problem. Fortunately, we can take the advantage that $f_V(s;\theta), f_{\mathcal{T}}(s,a;\phi), f_{\mathcal{R}}(s,a;\psi)$ are differentiable w.r.t the action vector $a$. We use Adam optimizer (Kingma & Ba, 2015) to solve the r.h.s optimization problem. To make sure that the optimizer finds a local maximum that is good enough to approximate the global maximum, we randomly select multiple initializations. Due to the superior parallel computing ability of modern GPUs, the optimization with different initial seeds can be operated simultaneously so that additional time consumption is limited. For exploration purpose, we also add noise (e.g. Ornstein-Uhlenbeck process) to the optimal action. While it is possible to expand the r.h.s. to a multi-step planning, notoriously, a learned model is inclined to diverge in long-horizon predictions. Consequently, we only use the one-step version for policy search.

To approximate the value function, we update the value approximator by the supervision of *temporal-difference error* (TD-error). Since updating the value function across the whole state space is unrealistic, we update the value approximator only by the data sampled from the environment in the current time step like policy gradient algorithms. Suppose at timestep $t$, the agent takes action $a_t$ (following Eq. 4), then receives reward $r_t$ and transits to state $s_{t+1}$. We minimize the following loss:

$$\mathcal{L}_{\mathcal{V}} = \|f_V(s_t;\theta) - (r_t + \gamma f_V(s_{t+1};\theta^-))\|^2. \tag{5}$$

Online training makes the whole framework more light-weight while sacrificing the guarantee of convergence. One can improve the training paradigm with past experience from the replay buffer or advanced sampling skills on the state space, but we leave them as future work.

Like DDPG, we employ a target network parameterized by $\theta^-$ to stabilize the training. To speed up computation, we also fork multiple agents to run in parallel and synchronize them by a global agent. Algotihm 1 recaptures the whole algorithm paradigm.

## 5 A Theoretical Justification from the Optimization Perspective

We will show the limitation of policy-centric methods and the nice properties of value-centric method. We narrow the analysis of policy-centric frameworks down to the landscape of $J(\pi)$. To save space, we leave the theorems and detailed proofs in Appendix A. We explain the intuitions here:

First, we show in Theorem 1 that, for an arbitrary MDP with a deterministic transition function, a local optimum of $J(\pi)$ that traps gradient-based methods could exist under a weak condition. In fact,

the local optimum issue is introduced by the parameterization of $\pi$'s output space, e.g., a Gaussian distribution. If we allow $\pi$ to be an arbitrary distribution, the local optimum will vanish (Proposition 1). The example in Sec 3 exactly shows a failure case of transferring policy due to a local optimum.

Second, we show in Theorem 2 that, for the same type of MDPs, the Continuous Value Iteration algorithm leveraging Property 1 in Sec 2.1 can share the favorable converge property of classical value iteration. That is, the distance between a current value function and the optimal value function will be squeezed at linear rate, thus it always converges. In addition, Proposition 2 implies that a small perturbation to the environment is only likely to cause a marginal change to the optimal value function; therefore, the old value function would serve as a good initialization point.

## 6 EXPERIMENTS

We first compare our value-centric method against the policy gradient baselines in the training from scratch setting. We also show that our value-centric method beats baselines in the transfer learning setting. Finally, we conduct ablation study over method parameters and diagnose the components.

### 6.1 SETUP

We evaluate our algorithm and two prevalent continuous RL methods, Deep Deterministic Policy Gradient (DDPG) (Lillicrap et al., 2015) and Trust Region Policy Optimization (TRPO) (Schulman et al., 2015), on five control tasks in the OpenAI Gym (Brockman et al., 2016):HalfCheetah-v1,InvertedPendulum-v1, InvertedDoublePendulum-v1, Pendulum-v0, and Reacher-v1. For baselines, we use the code from OpenAI Baselines (Dhariwal et al., 2017) and fine-tuned their hyper-parameters as much as we can. More details can be found in the Appendix D.

### 6.2 MAIN RESULTS

We compare the sample complexity and the performance of the three algorithms (MVC, DDPG, and TRPO) on the aforementioned five environments, for both the training from scratch setting and transfer learning setting. We also compare with PPO (Schulman et al., 2017) and SQL (Haarnoja et al., 2017) on HalfCheetah and show the results in Appendix F.

**Train from Scratch**    The reported results (Fig. 8(a)) show the mean and standard deviation of 3 runs with different random seeds. In four of the environments (InvertedPendulum-v1, HalfCheetah-v1, Pendulum-v0, Reacher-v1), our method achieves comparable performance as the better baseline. In InvertedDoublePendulum-v1, though there is a significant gap between MVC and TRPO, MVC performs at the same level with DDPG.

**Transfer across environments**    We demonstrate the superiority of our method when transferring across environments. For each of the above five environments, we change one or several physical properties, like the mass of pendulum, to create two new environments with novel dynamics. In one of the new environments, the change is relatively small (we call it 'Hard' in the plot), while the other is more intensively perturbed (we call it 'Harder' in the plot). We first train standard agents on the original environment. For fair comparison, we pick the agents that achieve comparable performance for all methods in the original environment. Please refer to Appendix D for the details of the modification and the agents. To avoid the possibility of under-exploration in the new environments, we reset the exploration noise of all the algorithms. We directly fine-tune all the agents with the same number of simulation steps. The results are shown in Fig. 8(b).

On all the environments, we observed that TRPO has the worst overall transfer performance. DDPG and MVC have similar transferrability on simple tasks like Reacher-v1 and Pendulum-v0. However, on more complicated tasks like HalfCheetah-v1,the performance of DDPG is significantly worse than MVC. Further investigation shows that DDPG can actually learn a high-quality $Q$ function for simple environments, which serves a similar role as our value function. However, on more challenging games such as HalfCheetah-v1 and InvertedDoublePendulum-v1, as a policy-centric algorithm, the learned $Q$ function is far from the true one (Fig. 7 in Appendix E), thus the transfer is significantly slower. The success and failure of DDPG again shows the central role value plays in transfer learning.

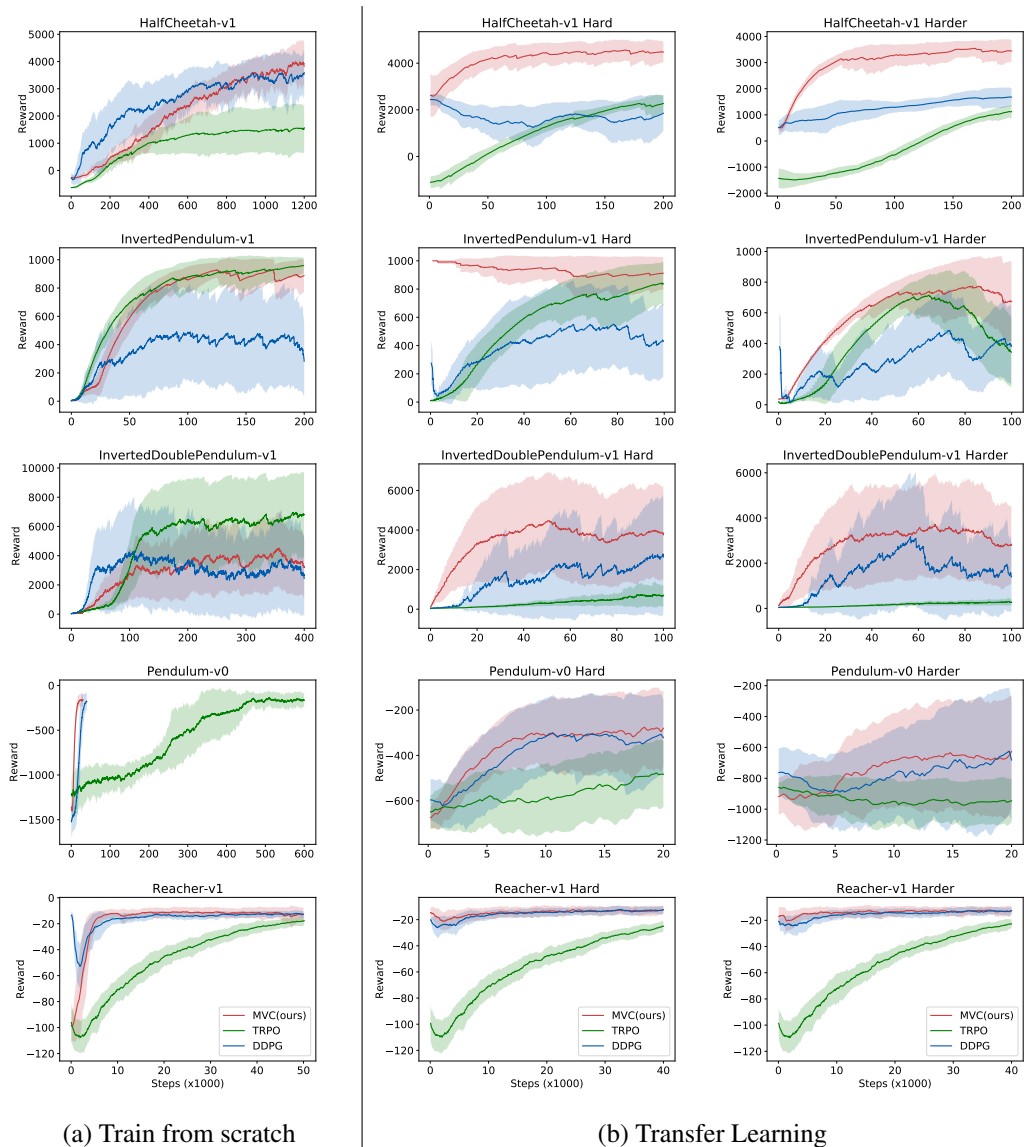

(a) Train from scratch                    (b) Transfer Learning

Figure 3: The training curves of training from scratch and transfer learning of MVC, TRPO, and DDPG. Thick lines correspond to mean episode rewards, and shaded regions show standard deviations of 3 random seeds. Our method (MVC) achieves comparable performance with the baselines while significantly outperforms them on transfer learning.

Note that, in HalfCheetah-v1-Harder, MVC achieves 3000 points in about 50k steps, while TRPO and DDPG only get around 1000 points after 200k steps.

## 6.3 ABLATION STUDY AND DIAGNOSIS OF COMPONENTS

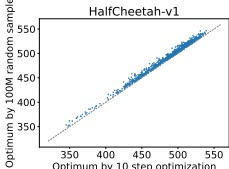

**Validation of Property 1**  Empirically, we find evidence for Property 1 and 2. Take HalfCheetah-v1 for example. For Property 1, we compare the optimization result of gradient-based method against random sampling with $10^6$ points.  Figure on the left demonstrates that Adam achieves comparable results with random sampling while being tens of times faster on our computer (a 48-core Xeon CPU with 2 Titan XP GPUs).

**Validation for Property 2** Figure on the right shows that the loss functions for the transition network and reward network converge in less than 100k time steps, which means the transition network and reward network converges much faster than the value network (As shown in Fig. 8(a), the value network still does not converge after 1200k steps.). Therefore, selecting actions based on the learned transition network and reward network is trusty. We also observed that the learned transition and reward networks provide good start points for the training in the new environment.

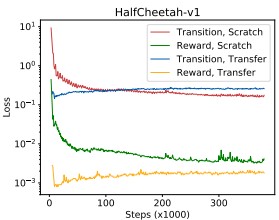

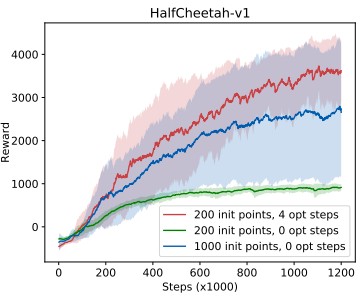

Figure 4: Train from scratch

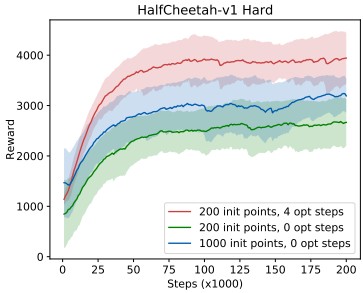

Figure 5: Transfer learning

**Hyperparameters** We verify the influence of the initialization and the number of optimization steps in the action search process. Fig. 4 shows that calling Adam optimizer is critical to stabilize the training. In transfer learning scenario (Fig. 5), the optimization shows a more significant impact on the performance and sample complexity. The agent that searches action with 200 initial points and 4 optimization steps is enough to win the agent using only 1000 initial points by a large margin.

## 7 RELATED WORK

**Reinforcement Learning in Continuous Control Domain.** Solving continuous control problems through reinforcement learning has been studied for decades (Sutton et al., 1998; Williams, 1992). Policy-based methods (Schulman et al., 2015; Mnih et al., 2016; Lillicrap et al., 2015) are more widely used. One exception is the NAF (Gu et al., 2016) method under the Q-learning framework which models the action space as a quadratic function.

**Value-based Reinforcement Learning.** The most relevant work in literature are perhaps the very recent Value Iteration Network (VIN) and Value Prediction Network (VPN) (Tamar et al., 2016; Oh et al., 2017). Though demonstrated better environment generalizability, VIN is specifically designed and only evaluated on the 2D navigation problem. VPN learns a dynamics model together with a value function and makes plans based on Monte Carlo tree search. In contrast to our work, VPN neither considered the continuous control problem nor thoroughly investigated their algorithm under the transfer learning setting.

**Model-based Reinforcement Learning.** For purposes such as increasing sample efficiency and designing smarter exploration strategies (e.g., curiosity-driven exploration), it can be beneficial to incorporate a learned dynamics model. Some very recent works have demonstrated the power of such model-based RL algorithms (Levine & Koltun, 2013; Nagabandi et al., 2017; Kurutach et al., 2018; Pathak et al., 2018; Feinberg et al., 2018; Pathak et al., 2017). However, to our knowledge, none of them has yet combined the value function with a learned dynamics model to solve continuous decision making problems.

**Transfer Learning in Deep Reinforcement Learning.** In this work, we study knowledge transfer problem across different MDPs.(Kansky et al., 2017) proposed the SchemaNetwork which learns the knowledge of the Atari physics engine by playing a standard version of the BreakOut game. (Higgins et al., 2017) learns disentangled representations in the source domains to achieve zero-shot domain adaption in the new environments. Finally, a straight-forward strategy is to show the agent all possible environments (Yu et al., 2017; Tan et al., 2018; Tobin et al., 2017).

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

## APPENDIX A    PROOFS FOR THE THEOREMS

For mathematical rigor, we suppose *the state space and the action space are both continuous in this section*.

### A.1    PROPOSITION 1 AND PROOF

**Proposition 1.** *If $J$ is the expected return defined as in Section 2, $\pi$ can be an arbitrary function defined on the probability simplex ($\int \pi(\tau)dx = 1$, $\pi(\tau) \geq 0$, where $\tau$ is a trajectory from the starting state to a terminal state), and $\pi^*$ is the maximizer of $J(\pi)$, then $\forall \pi^0$, there exists a path connecting $\pi^0$ to $\pi^*$ that $J(\pi)$ monotonically increases, which implies that $\pi^0$ is not a local maximum of $J$.*

*Proof.* Note that $J$ is a linear functional defined as $J(\pi) = \int \pi(\tau)R(\tau)$, where $R(\tau)$ is the total return following the trajectory $\tau$. Due to the linearity, we can generate the path by a convex interpolation of $\pi^0$ and $\pi^*$. Note that $J(\pi)$ is also a convex interpolation of $J(\pi^0)$ and $J(\pi^*)$ now. □

### A.2    PROPOSITION 2 AND PROOF

**Definition.** *Assume the transition probability $p(s^{'}|s, a)$ is perturbed by a disturbance $d(s^{'}|s, a)$, we denote the perturbed transition probability as $p_d(s^{'}|s, a) = p(s^{'}|s, a) + d(s^{'}|s, a)$. $d(s^{'}|s, a)$ is such a function that ensures $p_d(s^{'}|s, a)$ is a feasible distribution ($p_d(s^{'}|s, a) \geq 0$ and $\int_{s^{'}} (p_d(s^{'}|s, a)) = 1$). Define $||d|| = \max_{s \in \mathcal{S}, a \in \mathcal{A}} \int_{s^{'}} |d(s^{'}|s, a)|ds^{'}$.*

**Proposition 2.** *We assume $r(s, a)$ is bounded and denote $\max_{s,a} r(s, a) = R$. Denote the optimal value function for state $s$ in the perturbed environment is $V_d^*(s)$, and the optimal value of $s$ in the original environment is $V_0^*(s)$, then $|V_d^*(s) - V_0^*(s)| < \frac{R\epsilon}{(1-\gamma)^2}$ when $||d|| < \epsilon$ for $\forall s \in \mathcal{S}$*

*Proof.* We imitate the techniques in (Abbad & Filar, 1992). From the Markovian property, we have $V^*(s) = \max_a \int_{s^{'}} p(s^{'}|s, a)(r(s, a) + \gamma V^*(s^{'}))$

$$|V_d^*(s) - V_0^*(s)| \leq \max_a | \int_{s^{'}} ((p_d(s^{'}) - p_0(s^{'}))(r(s, a)) + \gamma(p_d(s^{'})V_d^*(s^{'}) - p_0(s^{'})V_0^*(s^{'})))|$$

$$\leq ||d||R + \gamma \max_a \int_{s^{'}} |d(s^{'})V_d^*(s^{'})|ds^{'} + \gamma \max_s |V_d^*(s) - V_0^*(s)|$$

Because $V^*(s) \leq \sum_{t=0}^{\infty} \gamma^t R = \frac{1}{1-\gamma}R$, we have,

$$max|V_d^*(s) - V_0^*(s)| \leq \frac{R||d||}{(1-\gamma)^2}$$

□

This bound is rather loose, and it can only be a partial explanation of the experimental results. The changes of environment in our experiments are much larger than a 'perturbation'.

### A.3    THEOREM 1 AND PROOF

**Definition.** *At the $i$-th iteration, we update the value function by the following rule:*

$$V^{i+1}(s) = (1 - \alpha)V^i(s) + \alpha \max_a [\mathcal{R}(s, a, \mathcal{T}(s, a)) + \gamma V^i(\mathcal{T}(s, a))], \forall s \in \mathcal{S} \tag{6}$$

*where $\alpha \in (0, 1]$ is the step size, and $V^0(s)$ is some initialization value. Note that by Property 1 in Sec 2.1, the maximization algorithm over $a$ is accessible.*

**Theorem 1.** *The above continuous value iteration algorithm is a contraction mapping:*

$$\max_s |V^k(s) - V^*(s)| \leq (1 - (1-\gamma)\alpha)^k \max_s |V^0(s) - V^*(s)|,$$

*where V\* represents the optimal value function.*

The proof is a trivial extension of conventional value iteration.

In addition, we analyze how the error of model approximators influence our algorithm. We suppose the error will not change when training the value function. Denote the dynamics approximator as $\hat{\mathcal{T}}(s, a)$ and the reward approximator as $\hat{\mathcal{R}}(s, a)$. Suppose the maximal error of the two approximators are respectively $\epsilon_T, \epsilon_R$, that is, $\max_{s,a} |\hat{\mathcal{T}}(s, a) - \mathcal{T}(s, a)| = \epsilon_T$ and $\max_{s,a} |\hat{\mathcal{R}}(s, a) - \mathcal{R}(s, a)| = \epsilon_R$. We have,

$$|V^{i+1}(s) - V^i(s)| \leq (1 - \alpha)|V^i(s) - V^{i-1}(s)| + \alpha\gamma \max_a |V^i(\hat{\mathcal{T}}(s, a)) - V^{i-1}(\hat{\mathcal{T}}(s, a))|, \forall s \in \mathcal{S}$$

As long as $\hat{\mathcal{T}}(s, a) \in \mathcal{S}$,

$$|V^{i+1}(s) - V^i(s)| \leq (1 - \alpha + \alpha\gamma) \max_s |V^i(s) - V^{i-1}(s)|, \forall s \in \mathcal{S}$$

Thus, the error will not hurt the contraction property, and the convergence is then guaranteed. However, the optimum may change. For the groundtruth dynamics and reward, the Bellman equation for the optimum is,

$$V^*(s) = \max_a [\mathcal{R}(s, a, \mathcal{T}(s, a)) + \gamma V^*(\mathcal{T}(s, a))], \forall s \in \mathcal{S}$$

For the approximated dynamics and reward,

$$V^*(s) = \max_a [\hat{\mathcal{R}}(s, a, \hat{\mathcal{T}}(s, a)) + \gamma V^*(\hat{\mathcal{T}}(s, a))], \forall s \in \mathcal{S}$$

They are not necessarily the same, hence the optimum may change.

### A.4 THEOREM 2 AND PROOF

**Lemma 1.** $\int_0^t e^{-x^2} dx > \frac{\sqrt{\pi}}{2}\sqrt{1 - e^{-\sqrt{t}}}, t > 0$

*Proof.* Denote the l.h.s. as A, then we have

$$A^2 = \int_0^t \int_0^t e^{-(x^2+y^2)} dxdy.$$

The integral is taken over a square area. By dividing the square area into a semi-circle area $D$ and the rest, we can derive the following inequality from the non-negativity of $e^{-(x^2+y^2)}$:

$$A^2 > \iint_D e^{-(x^2+y^2)} dxdy \quad = \int_0^{\frac{\pi}{2}} \int_0^t e^{-r^2} rdrd\theta$$

The r.h.s can be easily integrated, and the result is $\frac{\pi}{4}(1 - e^{-\sqrt{t}})$. Hence we get Lemma 1. $\qquad\square$

**Theorem 2.** *Denote the policy for a state $s$ as $\pi(s)$. For a given MDP, suppose*

$$\pi_0(s) = \begin{cases} \mathcal{N}(\mu_0, \sigma_0), s = s_0 \\ \pi^*(s), other \quad s \in \mathcal{S} \end{cases},$$

*where $\mu_0, \sigma_0$ are the initial parameters for the initial Gaussian policy $\pi_0(s_0)$, and all the other policies are fixed as the optimal policy $\pi^*(s)$. We simply denote such policy as $\pi(\mu_0, \sigma_0)$. Presume the optimal policy for $s_0$ is $\mathcal{N}(\mu^*, 0)$. If there exists $\mu \in [\mu_0, \mu^*]$ and $\Delta > 0$ such that the optimal Q value $Q^*(s_0, a) < B(\Delta)$ (B is a function w.r.t. $\Delta$) for all action $a \in [\mu' - \Delta, \mu' + \Delta]$. We have the following conclusion: $J(\pi(\mu', \sigma)) < J(\pi(\mu_0, \sigma_0))$ for all $\sigma \in (0, \sigma_0]$.*

*Proof.* We start the proof by showing the exact value of the bound $B$, then we demonstrate how we can deduce the conclusion from $B$. to simplify the proof, we use the following notations:

$$Q^*(s_0) = \max_a Q^*(s_0, a)$$

$$f(\mu, \sigma) = pdf(\mathcal{N}(\mu, \sigma))$$

$$V_\pi(s_0|\mu, \sigma) = \int_{-\infty}^{\infty} f(\mu, \sigma) Q_\pi(s_0, a) da$$

Note that since all the policies are fixed except for $s_0$, the $(\mu, \sigma)$ pair determines the whole policy $\pi$. Now using the above notations,

$$B = \frac{V_{\pi_0}(s_0|\mu_0, \sigma_0) - Q^*(s_0)}{\sqrt{1 - e^{-\sqrt{\frac{\Delta}{\sqrt{2}\sigma_0}}}}} + Q^*(s_0)$$

Recall the condition that for $a \in [\mu' - \Delta, \mu' + \Delta]$, $Q^*(s_0, a) < B$, we have:

$$V_{\pi'}(s_0|\mu', \sigma) = \int_{-\infty}^{\infty} f(\mu', \sigma) Q_{\pi'}(s_0, a) da$$

$$= \int_{-\infty}^{\mu' - \Delta} f(\mu', \sigma) Q_{\pi'}(s_0, a) da + \int_{\mu' - \Delta}^{\mu' + \Delta} f(\mu', \sigma) Q_{\pi'}(s_0, a) da + \int_{\mu' + \Delta}^{\infty} f(\mu', \sigma) Q_{\pi'}(s_0, a) da$$

Because $Q^*(s_0) \geq Q^*(s_0, a) \geq Q_{\pi'}(s_0, a)$,

$$V_{\pi'}(s_0|\mu', \sigma) \leq \int_{-\infty}^{\mu' - \Delta} f(\mu', \sigma) Q^*(s_0) da + \int_{\mu' - \Delta}^{\mu' + \Delta} f(\mu', \sigma) B \, da + \int_{\mu' + \Delta}^{\infty} f(\mu', \sigma) Q^*(s_0) da$$

$$= Q^*(s_0) + \int_{\mu' - \Delta}^{\mu' + \Delta} f(\mu', \sigma)(B - Q^*(s_0)) da$$

$$= Q^*(s_0) + (B - Q^*(s_0)) \int_{\mu' - \Delta}^{\mu' + \Delta} f(\mu', \sigma) da$$

From the following inequality and Lemma 1,

$$\int_{\mu' - \Delta}^{\mu' + \Delta} f(\mu', \sigma) da = \frac{1}{\sqrt{2\pi}\sigma} \int_{\mu' - \Delta}^{\mu' + \Delta} e^{-\frac{(a - \mu')^2}{2\sigma^2}} da$$

$$= \frac{1}{\sqrt{2\pi}\sigma} \int_{-\Delta}^{\Delta} e^{-\frac{a^2}{2\sigma^2}} da$$

$$\geq \sqrt{1 - e^{-\sqrt{\frac{\Delta}{\sqrt{2}\sigma}}}} \quad (Lemma 1)$$

$$\geq \sqrt{1 - e^{-\sqrt{\frac{\Delta}{\sqrt{2}\sigma_0}}}} \quad (\sigma \in (0, \sigma_0])$$

Thus, due to $B - Q^*(s_0) < 0$

$$V_{\pi'}(s_0|\mu', \sigma) \leq Q^*(s_0) + (B - Q^*(s_0))\sqrt{1 - e^{-\sqrt{\frac{\Delta}{\sqrt{2}\sigma_0}}}}$$
$$= V_{\pi_0}(s_0|\mu_0, \sigma_0)$$

Then, we can divide the expected return into two parts: 1. return from the paths that never pass $s_0$; 2. return from the paths that pass $s_0$. Since all the policies except for $s_0$ are deterministic, we can rewrite the expected return $J(\pi(\mu, \sigma))$ as:

$$J(\pi(\mu, \sigma)) = \alpha + \beta V_{\pi}(s_0|\mu, \sigma),$$

where $\alpha$ and $\beta$ are constants. Thus we finish our proof. $\qquad\square$

In fact, the bound $B$ is not hard to reach, a special case of the above theorem is ,

**Proposition.** *We follow the notations in Theorem 2, then, If $V_{\pi(\mu_0, \sigma_0)}(s_0) \geq 0.5 V_{\pi(\mu^*, 0)}(s_0)$, and $Q^*(s_0, a) < 0.6 V_{\pi(\mu_0, \sigma_0)}(s_0)$ for all $a \in [\frac{\mu_0 + \mu^* - \sigma_0}{2}, \frac{\mu_0 + \mu^* + \sigma_0}{2}]$, then such $\mu' \in (\mu_0, \mu^*)$ exists that $J(\pi(\mu', \sigma)) < J(\pi(\mu_0, \sigma_0))$ for all $\sigma \in [0, \sigma_0]$.*

The above proposition suggests that if the expected value of the original policy is not so bad $(V_{\pi(\mu_0, \sigma_0)}(s_0) \geq 0.5 V_{\pi(\mu^*, 0)}(s_0))$, a local minimum will emerge when the optimal Q of a part of the intermediate action is only a little lower than the initial value $(Q^*(s_0, a) < 0.6 V_{\pi(\mu_0, \sigma_0)}(s_0))$.

Although Theorem 1 proves the case when only the policy at $s_0$ can be optimized, it is clear that the result also holds if the whole policy can be optimized.

## APPENDIX B    DETAILS OF ILLUSTRATIVE EXAMPLE

We implement the MDP and the algorithms in MATLAB 2015b. For the policy gradient theorem, we use a variance-reduced version of the vanilla policy gradient:

$$\nabla_\theta J(\pi_\theta) = \frac{1}{N} \sum_{i=1}^{N} \sum_{t=1}^{T} \nabla_\theta \log \pi_\theta(a_t|s_t) (\sum_{t'=t}^{T} r(s_{t'}, a_{t'}))$$

For PG, we maintain a Gaussian policy $\mathcal{N}(\mu(s_i), \sigma^2(s_i))$ for $s_0, s_1, s_2$ with trainable parameters $\mu(s_i)$ and $\sigma(s_i)$ ($i = 0, 1, 2$). For transfer learning, in the new environment, we set the initial $\mu$s as the optimal policy in the source domain, and set $\sigma$s as 1 to encourage exploration. Likewise, for VI, we set the initial value function as the optimal value in the source domain.

For the value iteration, we use the continuous version as Section. 4. We set the step size of the policy gradient algorithm as 1e-4, and the step size of the value iteration algorithm as 1e-3.

Fig. 6 and table 1 give more details about the illustrative example.

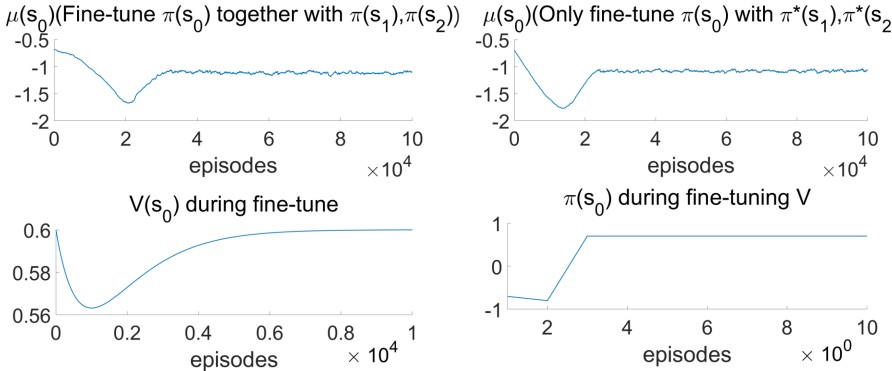

Figure 6: The training curve during fine-tuning.

|  | $s_0$ | $s_1$ | $s_2$ |
|---|---|---|---|
| $\mathbf{V}^*$ | 0.6 | 0.3 | 0.2 |
| $\mathbf{a}^*$ | -0.7 | -0.7 | 0.8 |

|  | $s_0$ | $s_1$ | $s_2$ |
|---|---|---|---|
| $\mathbf{V}^*$ | 0.6 | 0.2 | 0.3 |
| $\mathbf{a}^*$ | 0.7 | -0.8 | 0.7 |

Table 1: The Optimal Value and Policy in the Source Domain and the Target Domain (Left: Source Domain; Right: Target Domain)

## APPENDIX C    IMPLEMENTATION DETAILS

We use feed forward networks for all models and the activation function is ReLU. For transition network, reward network, and termination network, the networks have 2 hidden layers of size 512. For value network and target value network, the networks have 2 hidden layers of size 256 and 2 hidden layers of size 128. The weights of the networks are initialized by Xavier weight initializer. And the networks are all trained by Adam optimizer, the learning rates for transition network, reward network, termination network and value network are 1e-3, 1e-3, 1e-3, 1e-4, respectively.

The transition network, reward network, and termination network are trained 2 times for every 40 steps. The training data is uniformly sampled from the replay buffer, and the batch size is 512. The value network is trained 2 times in each step with the current $(s, r, s')$ tuple. At each time the value network is trained, the target networks will be soft updated as follows,

$$\theta^{Vtarget} = \tau\theta^V + (1 - \tau)\theta^{Vtarget} \tag{7}$$

A critical step in our algorithm is to find the optimal action $a^*$ with respect to state $s$. That is, we need to solve the following optimization problem,

$$\pi(s_t) = \arg\max_a (f_\mathcal{R}(s_t, a) + \gamma f_V(f_\mathcal{T}(s_t, a) + s_t)) \tag{8}$$

We use Adam optimizer with the learning rate of 0.1. To avoid trapped by local maxima, we set 200 initial points for the gradient descent procedure. When training from scratch, the optimizer will be called $k$ times, where the initial value of $k$ is 0 and it is increased by 1 for every 150k steps. The reason we do not optimize too many steps at the beginning is the value network is not accurate at that time. But when transferring, we set a fixed $k = 10$ because we have a good initialization for value network. After we get the action, a noise will be added to the action in order to encourage exploration. The noise we used is Ornstein-Uhlenbeck process with $\mu = 0$ and $\sigma = 0.15$. And the noise will be decayed with the progress of training.

However, the above case is just for the tasks without true terminations. For the tasks with true terminations, i.e. the task will be terminated before reaching the time limit, the optimization needs to be re-written as follows,

$$\pi(s_t) = \arg\max_a (f_\mathcal{R}(s_t, a) + \mathbb{1}_{f_\mathcal{D} < t}[\gamma f_V(f_\mathcal{T}(s_t, a) + s_t)]) \tag{9}$$

, where $t$ is the decision boundary of the termination network. Note that the above function is not fully differentiable, so we replace the hard threshold with a soft threshold.

$$\pi(s_t) = \arg\max_a (f_\mathcal{R}(s_t, a) + \sigma(t - f_\mathcal{D})(\gamma f_V(f_\mathcal{T}(s_t, a) + s_t))) \tag{10}$$

, where $\sigma$ is the sigmoid function. In this way, we make it fully differentiable, so that gradient descent can be applied.

Our implementation is based on PyTorch. In order to speed up our program, we employ multiprocessing technique. In practice, we have 4 processes running in parallel, and all the networks are shared across the processes.

## APPENDIX D    EXPERIMENT DETAILS

Our simulation environment is OpenAI gym 0.9.3.

For transferring experiments, we change one or several physical properties in the environments. In one of the new environments,the change is relatively small (we call it "Hard"), while the other is more intensively perturbed(we call it "Harder"). We achieve it by directly modifying the codes or the XML files in the OpenAI gym library. The modifications are listed in the Table 2.

|  | Hard | Harder |
|---|---|---|
| Pendulum-v0 | Change the mass from 1 to 2 | Change the mass from 1 to 2
Change the length from 1 to 2 |
| Reacher-v1 | Change the density of "body1" from 1000 to 2000 | Change the density of "body1" from 1000 to 2000
Change the damping of "joint1" from 1 to 2 |
| Inverted Pendulum-v1 | Change the gravity from -9.81 to -15 | Change the gravity from -9.81 to -20
Change the damping of all joints from 1 to 2 |
| InvertedDouble Pendulum-v1 | Change the gravity from -9.81 to -15 | Change the gravity from -9.81 to -20
Change the damping of all joints from 0.05 to 0.1 |
| HalfCheetah-v1 | Change the gravity from -9.81 to -15 | Change the gravity from -9.81 to -20
Change the damping of joint "bthigh" from 6 to 12
Change the damping of joint "bshin" from 4.5 to 9
Change the damping of joint "bfoot" from 3 to 6 |

Table 2: Environment modification

In transferring experiments, we select the agents get similar scores in the original environment for fair comparison. Table 3 shows the scores of the agents we chose in different environments. To get these scores, we evaluate each agent 10 times and calculate the mean episode reward.

|  | MVC(ours) | DDPG | TRPO |
|---|---|---|---|
| Pendulum-v0 | -163.1 | -124.7 | -110.9 |
| Reacher-v1 | -14.4 | -14.1 | -14.4 |
| InvertedPendulum-v1 | 1000.0 | 1000.0 | 1000.0 |
| InvertedDoublePendulum-v1 | 9354.3 | 9267.1 | 9112.4 |
| HalfCheetah-v1 | 3276.3 | 3391.6 | 4273.4 |

Table 3: Scores in the original environments.

## APPENDIX E  SUPPLEMENTARY MATERIALS FOR EXPERIMENTS

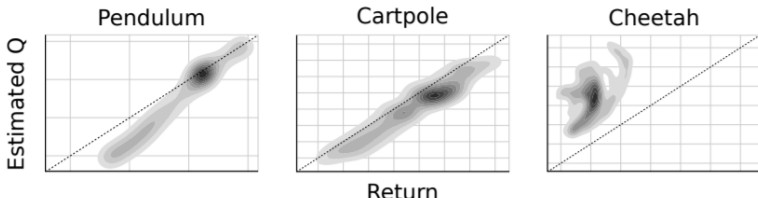

Figure 7: Density plot shows the estimated Q versus observed returns sampled from 5 test trajectories. for simple environments like Pendulum and Cartpole, the critic can predict the Q value quit accurate. However, in more complicated environment like Cheetah, the estimated Q are way more inaccurate. (This plot is from (Lillicrap et al., 2015))

## APPENDIX F  SUPPLEMENTARY EXPERIMENT RESULTS

We compare the sample complexity and the performance of the five algorithms (MVC, DDPG, TRPO, PPO, and SQL) on the HalfCheetah, for both the training from scratch setting and transfer learning setting.

SQL is a continuous variant of Q-learning, which expresses the optimal policy via a Boltzmann distribution. PPO is developed from TRPO with several modifications. In the train from scratch setting, SQL shows the best sample efficiency (significantly better than other algorithms). This is probably because it improves exploration by incorporating an entropy term but all the other algorithms do not. However, in the transfer learning setting, the performance of MVC is nearly the same as SQL. As both are value-centric algorithms, we think the benefit of MVC comes from using an environment dynamics approximator. PPO shows better performance than TRPO (both in the train from scratch and the transfer learning), but still much worse than MVC.

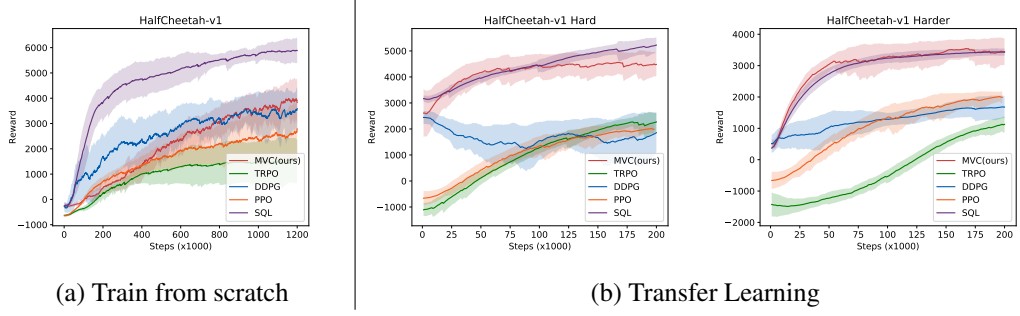

(a) Train from scratch                    (b) Transfer Learning

Figure 8: Comparison with more baselines.

We also did experiments with ME-TRPO, but we found it cannot handle a long time horizon (1000 time steps in HalfCheetah-v1) since its models will diverge quickly.

