# OpenReview forum: "Transfer Value or Policy? A Value-centric Framework Towards Transferrable Continuous Reinforcement Learning"
_ICLR.cc/2019/Conference_

### Official Review · AnonReviewer2 · 2018-11-02
**Has potential, needs some more investigation**

**Rating:** 5
**Confidence:** 2

**Review:**

This paper proposes a model-based value-centric (MVC) framework for transfer learning in continuous RL problems, and an algorithm within that framework. The paper attempts to answer two questions: (1) "why are current RL algorithms so inefficient in transfer learning" and (2) "what kind of RL algorithms could be friendly to transfer learning by nature"? I think these are very interesting questions to investigate, and researchers that work on transfer learning could benefit from insights on them. However, I am not yet convinced that this paper answers these questions satisfyingly. It would be great to hear the author's thoughts on my questions below.

The main insight I take away from the paper is that policy gradient methods are not suitable for transfer learning compared to model-based and value-centric methods for some assumptions (the reward function not changing and the transition dynamics being deterministic). This insight and the experiments in the paper are interesting, but I am unsure if the paper as it is presented now passes the bar for ICLR.

In general the paper has two contributions:
A) analysis of value-centric vs policy-centric methods
B) an algorithm that is more useful for transfer learning.

Regarding A)
The authors argue that policy-centric algorithms are less useful for transfer learning than value-centric methods.

They first illustrate this with an example in Section 3. Since this is just one example, as a reader I wonder if it would not be possible to construct an example that shows the exact opposite, where value iteration fails but policy gradient doesn't. It feels like there are many assumptions that play into the given example (the reward function not changing; the transition dynamics being deterministic; the choice of using policy gradients and value iteration).

In addition, the authors provide a theoretical justification in the Appendix (which I have briefly scanned) and the intuition behind it in Section 5. From what I understand, the main problem arises from the policy's output space being a Gaussian distribution, which causes the policy being able to get stuck in a local optimum. Further, the authors show (in the Appendix) that under some assumtions the value function always converges. Are there any guarantees on this when we don't have access to the true reward and transition functions (which themselves could get stuck in a local optimum)?

Would the authors say that the phenomenon is more a problem with the algorithm (policy gradient vs value iteration) than policy-centric and value-centric methods in general? Are there other methods that would be able to transfer policies better than policy gradient methods?

Regarding B)
The author's proposed method (MVC) has three components: the value function, the dynamics model and the reward model, all of which are learned by neural networks. It seems like the main advantage comes from using a model (since that's the aspect which changes when having to transfer to an altered MDP). Does the advantage of this method over DDPG and TRPO come from the fact that the dynamics model changes smoothly, and we have an approximation to it? Then it is not surprising that this outperforms a policy gradient method.

Other comments:

- Could you explain what is meant by "precise" and "imprecise" when speaking about policies or value functions?
- Could you explain what is meant by the algorithm being "accessible" (e.g., Definition 1)?

- Section 2.1: In Property 1, what is f? Could you make explicit why we are interested in the two properties listed? By "not rigorously", do you mean that those properties are based on intuition? These properties are used later in the paper and the appendix, so I wonder how strong of an assumption this is.
- Section 2.2: Could you explain what is meant by "task"? You say that within the MDP, the transition dynamics and reward functions change, but the task stays the same. However, earlier (in the introduction) you state that only the environment dynamics change. I find it confusing that "the task" is something hand-wavy and not part of the formal definition of the MDP. In what exact ways can the reward function be influenced by the change in the transition dynamics?
- Section 3: Replace "obviously" with "hence"; remove "it is not hard to find that". This might not be so trivial for some readers.
- Appendix B: Refer to Table 1 in the text.

Clarity: The paper is written well, but I think some assumptions and their affects should be stated more clearly and put into context. The paper misses a discussion / conclusion section. It would be great to see a discussion on some of the assumptions; e.g., what if the low dimensional assumtion breaks down? What if we assume that also the reward function can change? The authors are in a unique position to give insight into these things (even if the results from the paper do not hold after dropping some assumptions) and it would be very helpful to share these with the reader in a discussion section.

---

> ### Author Response · Authors · 2018-11-10
> **Response to Reviewer 2**
>
> We thank the reviewers for your appreciation of our work and valuable comments. We will clarify several misunderstandings and address your concerns.
>
> Our work is motivated by the observation that policy-centric methods are prone to get stuck in local minima in the transfer learning setting, as a small change to the dynamics may significantly change the optimal policy. Through theoretical, algorithmic, and experimental study, we show that our proposed value-centric method has better potential to transfer knowledge across environments in the setting that only dynamics changes but tasks keep the same.
>
> Q1: Is there a counterexample that fails value-based but not policy-based methods?
>
> A: The theory of value iteration (Theorem 1) guarantees that the algorithm always converges to the global optimum with linear rate (i.e., reach epsilon TD error in log(1/epsilon) steps). So the counterexample that traps value iteration in a local optimum does not exist.
>
> That said, in practice, value iteration may converge slower than (parameterized) policy-based methods in some cases. For example, if we change f1 = 0.7, f2 = 0.8 to f1 = 0.72, f2 = 0.78, then policy gradient may run faster with carefully chosen hyperparameters.
>
> Q2: Many assumptions for the example in Sec 3.
>
> A: These assumptions are reasonable for continuous control problem. That the transition function is deterministic is a common setting in continuous control problem. Also, the problem we study is that the task doesn't change so that we assume the reward function is not changed. Policy gradient is a mainstream continuous RL algorithm so we choose it for the comparison with our algorithm.
>
> Q3: Are there any guarantees on convergence when the true reward and transition functions are not accessible.
>
> A: The error with the reward/transition approximators does not affect the convergence property and rate. Simple analysis reveals that they may affect the optimum in theory, though. In practice, as our experiment in Sec 6.3 shows, the influence of errors from reward/transition approximators is basically negligible.
>
> Q4: Is the phenomenon more a problem with policy gradient than a general problem? Are there other methods that would be able to transfer policies better than policy gradient methods?
>
> A: We believe that the local minimum problem of transferring policy is a general challenge for all current RL algorithms *in the continuous setting*. Using terms defined in our paper, existing RL algorithms can be clustered as policy-centric and value-centric. We analyze the two types of algorithms as below.
>
> Policy-centric methods directly optimize the parameterized policy function. In the deep learning era, the majority of policy-centric methods are policy-gradient based. We have shown that two representative methods, TRPO and DDPG, perform worse than our method in the transfer learning setting.
>
> Value-centric methods make decisions based on a well-estimated parameterized value function. While classical value-centric methods (e.g., Q-learning and value iteration) are rarely used for continuous control problems, a variant of DQN called NAF [1] has been designed to deal with continuous action space. However, NAF uses a unimodal distribution to approximate the state-action value and it also suffers from the local optimum issue.
>
> Q5: Does the main advantage come from a smooth-changing model?
>
> A: Our advantage comes mainly from using the value function but not the model. As a comparison, even with a model, the policy still cannot be transferred efficiently. We elaborate this point as below:
>
> There are two mainstream methods for continuous RL algorithms that utilize models:
> 1. Emulate the real environment by a learned model and training a model-free RL agent in the emulated environment. In this case, note that policy gradient still cannot "jump" across remote actions. Actually, our toy example in Sec 3 has already revealed the limitation of this method. In the experiment, policy gradient method *converges* to a suboptimal local minimum even after abundant training, and note that the agent is training with access to the groundtruth environment model, even better than the learned model.
> 2. Model predictive control, which directly solves the action sequence {a_1, a_2, ... , a_T} according to the models. This is usually not feasible practically since it needs a very accurate model on the long horizon, while the approximated transition models learned by networks are only accurate on the short horizon.

---

> > ### Author Response · Authors · 2018-11-10
> > **(cont'd) Response to Reviewer 2**
> >
> > Q6: Explanation of some terminologies
> >
> > A:
> > 1. "Precise" and "imprecise" in Introduction: In the actor-critic framework, the actor (policy network) outputs a policy that leads to a high reward if the learning is successful. However, the critic module in practice will just produce a rather poor estimation of the true value for this policy, even after extensive training. This is what we mean by an 'imprecise' estimation. In fact, this phenomenon has been reported in [2] and inspired recent research over the problem [3]. We are happy to add references and further elaborate on this point in the paper.
> > 2. "Accessible" in Definition 1: We mean that we have access to an algorithm that is able to find the global maximum over the given formula. In practice, we use Adam initialized with multiple random seeds.
> >
> > Q7: Questions about Property1 and Property 2.
> >
> > A: The short answer is, the low-dimensional assumption is reasonable in continuous control, and its properties make our algorithm feasible.
> >
> > In Property 1,  f refers to the r.h.s. of Eq.4, and this property makes the optimization over Eq. 4 possible. Property 2 guarantees the function approximators in Eq.4 can be learned. The two properties are important to our algorithm and they do hold in practice.
> >
> > 'Not rigorously' means that the assumptions have been empirically validated in experiments, but we cannot show with mathematical rigor. We have demonstrated their correctness in the experiment section on HalfCheetah, which has the highest dimensions of state and action space among all five environments.
> >
> > Q8: The question of 'task' and the change of reward.
> >
> > A: How to define a task precisely is still an open problem in the control/RL community. But we can give some examples. In pendulum, the task is to swing the pendulum upright; in HalfCheetah, the task is to move forward as fast as possible.
> >
> > Here we explain why the reward r(s_t, a_t) may change. Take HalfCheetah as an example, the reward function is proportional to the distance between the position x of the cheetah in frame t and frame t+1, so actually the reward function is r(s_t, a_t, s_t+1) (and this function won't change). Because s_t+1 = T(s_t, a_t), we have r(s_t, a_t, s_t+1) = r(s_t, a_t, T(s_t, a_t)). It is clear that r(s_t, a_t) = r(s_t, a_t, T(s_t, a_t)) changes as T changes. However, the goal is still to move forward as fast as possible. We find that directly approximate r(s_t,a_t) works well in the experiments, so we did not take the following state s_t+1 into account. We did not explain the details in the paper due to space limitation.
> >
> > Q9: Discussion/ Conclusion.
> >
> > A: We did not include a discussion/conclusion section due to the limitation of space. Here we would like to discuss the concerns you raised.
> >
> > First, if the low-dimensional assumption breaks down, our method might fail. However, we envision that high-quality and low-dimensional environment models will be accessible as state abstraction techniques maturate, and our method will show even greater potential then. In fact, the investigation into compact model-based learning methods has started to be advocated by some pioneer researchers in the RL community. For example, Prof. Richard Sutton said in a recent talk at Alberta University that, 'one next big step in AI is planning with a learned model'.
> >
> > Second, if the goal of the task changes, while the value and the reward approximator will be useless, the learned transition approximator is still useful to accelerate the learning of new task.
> >
> > Finally, we want to emphasize again very few formal studies have been done in our transfer learning setting before, and our algorithm could be a novel perspective on the problem. And in { https://openreview.net/forum?id=H1gZV30qKQ&noteId=rJgTKis-6m }, we emphasize our contributions by positioning this work in the literature.
> >
> > References:
> > [1] Gu, Shixiang, et al. "Continuous deep q-learning with model-based acceleration." International Conference on Machine Learning. 2016.
> > [2] Lillicrap, Timothy P., et al. "Continuous control with deep reinforcement learning." arXiv preprint arXiv:1509.02971 (2015).
> > [3] Feinberg, Vladimir, et al. "Model-Based Value Estimation for Efficient Model-Free Reinforcement Learning." arXiv preprint arXiv:1803.00101 (2018).

---

> > > ### Comment · AnonReviewer2 · 2018-11-13
> > > **reply**
> > >
> > > Thank you for your thorough response and clarifications. This made some things clearer, and a few things could be updated in the paper. You can go above the 8 page limit (the recommended limit is 8, but the hard limit is 10) to add a conclusion.
> > >
> > > I still think this is an interesting line of work, but might need some more research/experiments to gain novel insights and convince the reader.

---

> > > > ### Author Response · Authors · 2018-11-27
> > > > **Revision Uploaded**
> > > >
> > > > Dear reviewer, we have updated our paper. To address your concern about the convergence without the true reward and transition function, we analyze how the error of model approximators influence our algorithm in Appendix A.3 (page 11-12). Please check our new version.

---

### Official Review · AnonReviewer1 · 2018-11-05
**Overall interesting, but concerns about the key idea and the applicability of the method**

**Rating:** 4
**Confidence:** 4

**Review:**

The paper considers the problem of transfer in continuous-action deep RL. In particular, the authors consider the setting where the dynamics of the task change slightly, but the effect on the policy is significant. They suggest that values are better suited for transfer and suggest learning a model to obtain these values.

Overall, there are interesting ideas here, but I am concerned about whether the proposed approach actually solves the problem the authors consider and its general applicability.

The point about value functions being better suited for transfer than policies is indeed true for greedy policies: it is well-known that they are discontinuous, and small differences in value can result in large differences in policy. This point is hence relevant in continuous control, where deterministic policies are considered.

But I am a bit confused as to why the proposed approach is better though. Eq. (4) still takes a max w.r.t. the estimated dynamics, etc. So even if the value function is continuous, by taking the max, we get a deterministic policy which has the same problem! That is probably why the performance is quite similar to DDPG. Considering a softer policy parameterization (a continuous softmax analogue) would be more in line with the authors’ motivation.

The proposed method itself doesn’t seem generally practical unfortunately, as it is suggested to learn the *model* of the environment for with a high-dimensional state space and a continuous action space, and do value iteration. In other words, if Property 2 was easy to satisfy, we wouldn’t be struggling with model-based methods as much as we are! However, I do appreciate that the authors illustrate the model loss curves in their considered domains. This raises a question of when are dynamics “easy”.

The theoretical justification is quite weak, since the bound in Proposition 2 is too loose to be meaningful (as the authors themselves acknowledge). One way to mitigate this would be to support it empirically, by considering a range of disturbances of the specified form, and showing the shape of the bound on a small domain. The same thing can be done for the parametric modifications considered in the experiments -- instead of considering a set of instances, consider the performance as a function of the range of disturbances to the same dynamics parameter.

Minor comments:
* The italicization of certain keywords in the intro is confusing, in particular precise, imprecise -- these aren’t well-defined terms, and don’t make sense to me in the mentioned context. The policy function isn’t more “precise” than the value.
* I suggest including the statements of the propositions in the main text

---

> ### Author Response · Authors · 2018-11-08
> **Response to Reviewer 1**
>
> Thanks for the valuable feedback! We would like to first address your concerns and then restate our key contributions that might not have been fully appreciated.
>
> Our work is motivated by the observation that policy-centric methods are prone to get stuck in local minima in the transfer learning setting, as a small change to the dynamics may significantly change the optimal policy. Through theoretical, algorithmic, and experimental study, we show that our proposed value-centric method has better potential to transfer knowledge across environments in the setting that only dynamics changes but tasks keep the same.
>
> Response to your concerns:
>
> Q1: By taking the max, the proposed method has no superiority to a deterministic policy.
>
> A: First of all, the max operator provides an advantage irrelevant to whether the policy is deterministic or not, as elaborated below. As shown in our example (Sec 3), value-centric methods allow updating the old action $a$ at some state to a new action $a'$ that can be *arbitrarily far away* in the action space in a single step. Algorithmically, this flexibility is achieved by the max operator -- as long as the new action $a'$ gives a better return than the old action $a$, $a'$ will be chosen in place of $a$. This behavior grants value-centric methods much stronger power compared with existing policy gradient-based methods in the transfer learning setting.
>
> To be even more specific, in continuous control, there are 2 major policy parameterizations:
> 1. A neural network outputs a deterministic policy, e.g. DDPG
> 2. A neural network outputs the parameters of a Gaussian distribution, then we sample an action from this distribution, e.g. TRPO, PPO and A3C
>
> To achieve swift adaptation, the old policy on a specific state may need to be updated to another remote action in the learning process (as in the example of Sec 3). Unfortunately, policy parameterizations listed above could only make local adjustment by updating along the gradient direction. They cannot "jump" across remote actions in a single step. In contrast, our method *derives* the policy from the value function. As discussed above, this derived policy could "jump" by nature.
>
> We are not sure what you mean by "a softer policy parameterization". If you mean a Gaussian policy, that is the Parameterization 2 above and it would not work (as shown by the TRPO baseline in many of our experiments). In fact, one possible way is to consider a multi-modality distribution, but we don't see such design in literature, possibly because it is very hard to implement in practice. Again, in contrast, our methods could derive a multi-modality policy by nature.
>
> Q2: The models are not easy to learn in the high-dimensional situation, and thus Property 2 is not easy to satisfy.
>
> A: First, it is generally accepted that models (especially one-step transition models) are not hard to learn in the continuous control domain (our topic, as in the paper title). This is already evidenced in our experiments (Sec 6.3, validation for Property 2). Additionally, existing published works [2] (NIPS) and [4] (ICLR) learned the environment models similarly as we do, and have already solved all control tasks in OpenAI Gym in the train-from-scratch setting. Therefore, learning the model in continuous control tasks is not a problem.
>
> Second, although learning models from high-dimensional visual input is indeed hard, the "struggle" is still likely to be indispensable towards the ultimate goal of sample-efficient planning. While in such tasks Property 2 is still not fully satisfied today, we envision that high-quality low-dimensional models of environments will be accessible as state abstraction techniques maturate, and our method will show even greater potential then. In fact, the investigation into model-based methods has started to be advocated by some pioneer researchers in the RL community. For example, Prof. Richard Sutton said in a recent talk at Alberta University that, 'one next big step in AI is planning with a learned model'. This year, model-based methods have already led to some astonishing results. [8, 4].
>
> Q3: Confusion over 'precise' and 'imprecise'.
>
> A: In an actor-critic framework, the actor (policy network) outputs a policy that leads to a high reward if the learning is successful. However, the critic module in practice will just produce a rather poor estimation of the true value for this policy, even after extensive training. This is what we mean by an 'imprecise' estimation. In fact, this phenomenon has been reported in [5] and inspired recent research over the problem [3]. We are happy to add references and further elaborate this point in the paper.

---

> > ### Author Response · Authors · 2018-11-08
> > **(cont'd) Response to Reviewer 1**
> >
> > Finally, we want to emphasize our contributions by positioning this work in the literature.
> >
> > While transfer learning in RL has been investigated in different settings, our setting (only dynamics changes but tasks keep the same across environments) has not been *formally* studied yet. Prior work such as [6, 7] mostly targets at relevant but different settings. In methodology, they either need additional knowledge of what environment parameters would change or rely on domain randomization, thus are quite sample inefficient. We believe that there must exist fundamentally different mechanisms that result in sample-efficient algorithms in our setting, as humans can adapt to environmental change quickly with neither explicit knowledge about the change nor domain randomization.
> >
> > We provide a fundamental and new perspective to this problem. We rethink what knowledge is really needed to adapt to the new environment. Our brand-new framework is based on the recognition that the knowledge of both environment dynamics and state values are indispensable during the transfer process.
> >
> > We firmly believe that we have two significant contributions:
> >   - Our algorithm, MVC, is the *first attempt* to address the problem in a value-centric way. MVC is not a trivial extension of value iteration since value iteration relies on ground truth dynamics and has not been applied to deal with continuous actions.
> >   - MVC also suggests that "directly search the action that gives the maximum value" could work if both state and action spaces are continuous, a point that has not been demonstrated before. As far as we know, MVC is the first continuous RL algorithm using this paradigm, though many previous works claim this paradigm "impossible" [1, 2, 3].
> >
> > We are looking forward to further discussions. Thank you!
> >
> > References:
> > [1] Gu, Shixiang, et al. "Continuous deep q-learning with model-based acceleration." International Conference on Machine Learning. 2016.
> > [2] Buckman, Jacob, et al. "Sample-Efficient Reinforcement Learning with Stochastic Ensemble Value Expansion." arXiv preprint arXiv:1807.01675 (2018).
> > [3] Feinberg, Vladimir, et al. "Model-Based Value Estimation for Efficient Model-Free Reinforcement Learning." arXiv preprint arXiv:1803.00101 (2018).
> > [4] Kurutach, Thanard, et al. "Model-Ensemble Trust-Region Policy Optimization." arXiv preprint arXiv:1802.10592 (2018).
> > [5] Lillicrap, Timothy P., et al. "Continuous control with deep reinforcement learning." arXiv preprint arXiv:1509.02971 (2015).
> > [6] Rajeswaran, Aravind, et al. "Epopt: Learning robust neural network policies using model ensembles." arXiv preprint arXiv:1610.01283 (2016).
> > [7] Yu, Wenhao, et al. "Preparing for the unknown: Learning a universal policy with online system identification." arXiv preprint arXiv:1702.02453(2017).
> > [8] Pathak, Deepak, et al. "Zero-shot visual imitation." International Conference on Learning Representations. 2018.

---

### Official Review · AnonReviewer4 · 2018-11-11
**Limited novelty and inconclusive experiments**

**Rating:** 5
**Confidence:** 3

**Review:**

The paper proposes a model-based value-centric (MVC) deep RL algorithm for transfer learning. The algorithm optimizes neural networks to estimate the deterministic transitions and rewards, and uses the these models to learn a value function by minimizing the Bellman residual. Policy is represented implicitly as the action that greedily maximizes the return, expressed in terms of the learned models. The experiments show some improvement on transferability over DDPG and TRPO policies.

The paper has two relatively independent stories: The title and the introduction motivates the work by discussing the transferability of policies and value functions. However, instead of rigorously evaluating transferability, the paper proposes a model-based algorithm (MVC) for learning policies for continuous actions. Novelty of the new algorithm is quite limited, as it simply uses a learned dynamics model and reward function to learn a value function. Regarding transferability, introducing MVC seem quite orthogonal, and instead, it would be better to have a clear comparison of transferability using existing methods (e.g., DDPG). If having an explicit policy network hurts transferability, then existing algorithms can be modified by replacing the actor with greedy maximization, or alternatively other value based methods that do not involve actor network (NAF, SQL, QT-Opt) could be used.

Regarding the intuition why values transfer better, the examples given in the introduction and Section 3 are good and intuitive. However, from my experience, the limited information content of a policy is only a partial reason for poor transferability, and in practice I have seen policies to transfer, in fact, better than values. The chosen viewpoint based on information content is nice as it can be proven mathematically, but might not be the most insightful and important in practice. The experimental evaluation is not rigorous enough to allow drawing further conclusions. For example, one could compare the two approaches using a wider set of RL algorithms, include more realistic environments (ideally transfer to real-world), or have a heat map illustrating transferability w.r.t. selected parameters. Also, no comparison to the  state-of-the-art methods is provided (PPO, TD3, SAC).

Minor points:
- Please include the theorems in Section 5 (and proofs in the appendix). The intuition provided in the body is not very clear.
- Why is it necessary to assume a deterministic dynamics model? Why only the dynamics model can vary between the domains and not also the reward (second paragraph in Section 1)?

---

> ### Author Response · Authors · 2018-11-14
> **Response to Reviewer 4**
>
> Thanks for the valuable feedback! We will address your concerns one by one.
>
> Q1: Lack of rigorous evaluation of transferability.
> A: There lacks a rigorous standard to measure the transferability of an RL algorithm. To evaluate the transferability empirically, we are working on adding more results of your proposed algorithms in the experiment section before the rebuttal deadline. Stay tuned.
>
> Q2: There are some possible solutions to the drawbacks of explicit policy, like replacing the actor with greedy maximization, or other value-based methods.
> A: If we replace the actor with greedy maximization, the algorithm will become a value-centric algorithm (not exactly ours), because it also directly optimizes the value function and derives the policy from value. We are working on the comparison with your proposed baseline before the rebuttal ends (e.g., comparison with state-of-the-art value-centric algorithms like SQL). For NAF, it uses a unimodal approximation of the Q function, and it will suffer from a similar issue like policy gradient methods.
>
> Q3: In practice, the reviewer has seen policies transfer better than values.
> A: Our point is that policy gradient methods may get stuck in certain situations, while in contrast, value-centric method enjoys the linear convergence property in theory. We don't mean policy gradient methods will necessarily get stuck every time.  In practice, policy gradient methods are possible to beat value-centric methods with careful hyperparameter tuning. Also, we are wondering what your experiment setting is when you observed policy transfers faster than value.
>
> Q4:  The experimental evaluation is not enough to allow drawing further conclusions.
>
> A: We are working on the experiments (e.g. a heat map illustrating transferability, comparison with PPO and SAC) and hopefully provide the results before the rebuttal deadline.
>
>
> Q5: Why is it necessary to assume a deterministic dynamics model? Why only the dynamics model can vary between the domains and not also the reward (second paragraph in Section 1)?
> A: It is not necessary to assume a deterministic dynamics model. We take the deterministic assumption just because of its simplicity and performance. Additionally, people use a deterministic transition approximator in continuous control extensively [1, 2, 3].
>
> We assume that the reward function would not change because it is the reward that defines the task. This aligns with the problem we study in the paper -- only the dynamics changes but the *task* keeps the same across environments.
>
> [1] Kurutach, Thanard, et al. "Model-Ensemble Trust-Region Policy Optimization." arXiv preprint arXiv:1802.10592 (2018).
> [2] Feinberg, Vladimir, et al. "Model-Based Value Estimation for Efficient Model-Free Reinforcement Learning." arXiv preprint arXiv:1803.00101 (2018).
> [3] Pathak, Deepak, et al. "Zero-shot visual imitation." International Conference on Learning Representations. 2018.

---

> > ### Author Response · Authors · 2018-11-27
> > **Revision Uploaded**
> >
> > Dear reviewer, we have updated our paper. Results of two new baselines on HalfCheetah-v1 have be added in Appendix F (page 16). Please check our new version.

---

> > > ### Comment · AnonReviewer4 · 2018-12-01
> > > **Response**
> > >
> > > I greatly appreciate including some of the requested comparisons in the appendix. The new results indicate that existing algorithms can match or even outperform MVC in both learning from scratch and in the transfer learning setup, and since MVC has limited novelty, I will stick with my earlier assessment. However, the results also indicate that methods that do not have an explicit actor (MVC and SQL) transfer better, and thus I would recommend revising the paper and making transferability the main contribution (instead of MVC), and resubmitting to a future conference.

---

### Author Response · Authors · 2018-11-27
**Revision Uploaded**

We thank the reviewers for their valuable comments.  Accordingly, we have updated our paper. The specific changes go as follows.

1. We analyze how the error of model approximators influence our algorithm in Appendix A.3 (page 11-12)
2. We compare with more baselines (PPO and SQL) on HalfCheetah and show the results in Appendix F (page 16). We revise the figure 2 in Section 3 to present the illustrative example in a more clear way.

Please check our new version.

---

### Meta-Review · Area_Chair1 · 2018-12-14
**The paper needs to be imrpvoed**

**Confidence:** 4
**Recommendation:** Reject

**Metareview:**

The paper studies whether the best strategy for transfer learning in RL is to transfer value estimates or policy probabilities. The paper also presents a model-based value-centric (MVC) framework for continuous RL. The reviewers raised concerns regarding (1) the coherence of the story, (2) the novelty and importance of the MVC framework and (3) the significance of the experiments. I encourage the authors to either focus on the algorithmic aspect or the transfer learning aspect and expand on the experimental results to make  them more convincing. I appreciate the changes made to improve the paper, but in its current form the paper is still below the acceptance threshold at ICLR.

PS: in my view one can think of value as (shifted and scaled) log of policy. Hence, it is a bit ambiguous to ask whether to transfer value or policy.